# Post-Transplant Diabetes Mellitus in Kidney-Transplanted Patients: Related Factors and Impact on Long-Term Outcome

**DOI:** 10.3390/nu16101520

**Published:** 2024-05-17

**Authors:** Carlo Alfieri, Edoardo Campioli, Paolo Fiorina, Emanuela Orsi, Valeria Grancini, Anna Regalia, Mariarosaria Campise, Simona Verdesca, Nicholas Walter Delfrate, Paolo Molinari, Anna Maria Pisacreta, Evaldo Favi, Piergiorgio Messa, Giuseppe Castellano

**Affiliations:** 1Nephrology, Dialysis and Transplantation, Fondazione IRCCS Ca’ Granda Ospedale Maggiore Policlinico, 20122 Milan, Italy; anna.regalia@policlinico.mi.it (A.R.); maria.campise@policlinico.mi.it (M.C.); simona.verdesca@policlinico.mi.it (S.V.); nicholas.delfrate@unimi.it (N.W.D.); paolo.molinari1@unimi.it (P.M.); anna.pisacreta@unimi.it (A.M.P.); piergiorgio.messa@unimi.it (P.M.); giuseppe.castellano@unimi.it (G.C.); 2Department of Clinical Sciences and Community Health, Università degli Studi di Milano, 20122 Milan, Italy; evaldo.favi@unimi.it; 3General Surgery and Kidney Transplantation, Fondazione IRCCS Ca’ Granda Ospedale Maggiore Policlinico, 20122 Milan, Italy; edoardo.campioli@unimi.it; 4Division of Endocrinology, ASST Fatebenefratelli-Sacco, 20122 Milan, Italy; paolo.fiorina@unimi.it; 5International Center for T1D, Pediatric Clinical Research Center Romeo et Enrica Invernizzi, Department of Biomedical and Clinical Science L. Sacco, University of Milan, 20122 Milan, Italy; 6Diabetes Unit, Foundation IRCCS Cà Granda Ospedale Maggiore Policlinico, 20122 Milan, Italy; emanuela.orsi@policlinico.mi.it (E.O.); valeria.grancini@policlinico.mi.it (V.G.); 7Post-Graduate School of Specialization in Nephrology, University of Milano, 20157 Milan, Italy

**Keywords:** kidney transplant, OGTT, post-transplant diabetes mellitus, graft failure, survival

## Abstract

This study aimed to investigate the prevalence and determinants of glucose metabolism abnormalities and their impact on long-term clinical outcomes in kidney transplant recipients (KTxps). A retrospective analysis of 832 KTxps (2004–2020) was performed. Patients were assessed at 1 (T1), 6 (T6), and 12 (T12) months post-transplantation and clinically followed for an average of 103 ± 60 months. At T6, 484 patients underwent an oral glucose tolerance test for the diagnosis of alterations in glucose metabolism (AMG+) or post-transplant diabetes mellitus (PTDM+). The prevalence of pre-transplant diabetes was 6.2%, with 22.4% of PTDM+ within the 1st year. Patients with AMG were older and exhibited altered lipid profiles, higher body mass index, and increased inflammatory indices. Age at transplantation, lipid profile, and inflammatory status were significant determinants of PTDM. Graft loss was unaffected by glucose metabolism alterations. Survival analysis demonstrated significantly worse long-term survival for KTxps with diabetes (pre- and PTDM+, *p* = 0.04). In a comparison of the ND and PTDM+ groups, no significant differences in death with a functioning graft were found. The AMG+ group exhibited worse survival (*p* < 0.001) than AMG−, even after excluding patients with diabetes mellitus. Future randomized controlled trials are necessary to delve deeper into this subject, specifically examining the effects of new antidiabetic treatments.

## 1. Introduction

Kidney transplantation (KTx) is the best therapy for the treatment of end-stage renal disease (ESRD) because it guarantees better patient survival and quality of life than chronic dialysis treatment. In kidney-transplanted patients (KTxps), alterations in glucose metabolism have a particularly high incidence. Numerous studies identify them as relevant causes of high cardiovascular mortality [1,2,3].

Post-transplant diabetes mellitus (PTDM) describes newly diagnosed diabetes mellitus after transplantation [4]. Hyperglycemia in the immediate post-KTx period is frequent, affecting about 90% of KTxps in the first weeks of KTx [5]. The cause can be related to antirejection therapies, infectious states, and other critical conditions. This type of glycemic disturbance is strongly associated with the subsequent development of PTDM. Consequently, to make a diagnosis of PTDM, it is advisable to wait for the stabilization of the patient’s immunosuppressive therapy and renal function and to ascertain the absence of concomitant infectious events, such as wounds or systemic bacterial, viral, or fungal infections [6].

According to the literature, the incidence of impaired glucose metabolism after KTx is around 10–40% [7]. This great variability is largely justified by the diversity of the diagnostic parameters used (e.g., use of antidiabetic drugs, glycemic values, glycated hemoglobin), by the different exclusion criteria, by the diversity of the therapeutic schemes adopted, and by the choice of follow-up time. The present retrospective observational monocentric study aimed to analyze the prevalence and determinants of PTDM and anomalies of glucose metabolism, and to assess their influence on the main long-term clinical outcomes in a cohort of KTxps.

## 2. Materials and Methods

### 2.1. Study Design

In the Nephrology, Dialysis and Kidney Transplantation Unit of the Foundation Ca’ Granda IRCCS Policlinico in Milan, more than 1600 KTxps are actually followed up, coming from the Lombardia region and the whole of Italy. In our Center, between January 2004 and October 2020, 879 patients underwent KTx. Of these patients, 29 did not reach the sixth month of follow-up, and 18 were lost during the global follow-up. All these patients were not considered in the study, and they did not differ from the studied cohort. Therefore, 832 patients were included in the present study.

The treatments and procedures reported herein were in accordance with the ethical standards of the institutional committee of the institution at which this study was conducted (Fondazione IRCCS Ca’ Granda Ospedale Maggiore Policlinico Ethical Committee, Protocol ID 4759-1837/19–11/2019), as well as with the 1964 Helsinki declaration and its later amendments, or comparable ethical standards. All participants consented for research purposes at the time of activation on the transplant waiting list, and their willingness to participate in future non-interventional research projects was confirmed before the transplant procedure. Donor data, organ details, recipient characteristics, and study-related outcomes were recorded in a central database by dedicated staff (as per local practice) and reviewed by the authors.

The cohort was divided into groups according to glucose metabolism parameters, as done in our previous study [8] and as reported in Figure 1 and Appendix A: non-diabetic patients (ND, n = 595), diabetics (diabetics before transplantation + new diabetics during the first year of transplantation, D, n = 237), pre-transplant diabetics (D1, n = 51), post-transplant diabetics (PTDM+, n = 186), patients with OGTT alterations (AMG+, n = 113), patients without OGTT alterations (AMG−, n = 371).

The 832 KTxps were studied to explore the relationships and impact of metabolic and KTx-related development of post-transplant glucose metabolism disarrangement. The analyses, in addition, would permit the assessment of post-transplant glucose metabolism disarrangement’s influence on the main long-term clinical outcomes. Briefly, the KTxps underwent the following:-Collection of pathological and pharmacological history and complete physical examination, at the time of inclusion in the study (T1);-Detection of clinical and biochemical parameters at 1, 6, and 12 months (T1, T6, and T12, respectively) after transplantation;-Specific biochemical determinations of glucose metabolism (glycemia, glycated hemoglobin, and insulin) and concerning mineral metabolism, in particular parathyroid hormone (PTH), calcium (Ca), phosphorus (P), and native and active vitamin D levels (25OH and 1-25OH) at T1, T6, and T12;-The collection of data concerning induction and maintenance of immunosuppressive therapy (at T1 and T12) and antihypertensive, antidiabetic, lipid, and vitamin D treatments, as well as the calculation of cumulative doses of steroid therapy at T1, T6, and T12;

At T6, execution and analysis of the oral glucose tolerance test (OGTT).

### 2.2. Measurements and Definitions

Patients underwent biochemical analysis after 12 h of fasting. Samples were taken at the transplant clinic of our department.

Concerning glucose homeostasis, at T1, T6, and T12, blood glucose, insulinemia, and glycated hemoglobin were measured.

Following the American Diabetes Association (ADA) criteria [9], the presence of PTDM was defined by the presence of the following: (a) fasting plasma glucose levels ≥126 mg/dL confirmed by repeat testing on a different day; (b) HbA1c levels permanently ≥6.5%; (c) absence of other related causes, such as stress hyperglycemia, pancreatitis, or use of diabetogenic medications other than immunosuppressive drugs.

At T6, in addition, 484 patients underwent OGTT, following the indications of the ADA [9]: after a night of fasting, and after an oral glucose load of 75 g, venous samples were collected for the dosage of blood glucose and insulinemia at 0, 30, 60 and 120 min. The examination was not performed in 51 patients because these patients were already diabetic before KTx. In addition, the examination was not carried out in 297 patients for clinical, organizational, or logistical reasons or because of the development of diabetes in the first 6 months post-transplant. A comparative analysis between patients who underwent OGTT and those who should have undergone OGTT revealed that patients who underwent OGTT had a longer duration of follow-up and a lower age at transplantation.

Considering the result of the OGTT, according to ADA definitions, patients were grouped into the following groups:-Normoglycemic patients (AMG−);-Patients with alterations in glucose metabolism (AMG+):-IFG (Impaired Fasting Glucose): fasting blood glucose between 100 and 125 mg/dL;-IGT (Impaired Glucose Tolerance): blood glucose between 140 and 199 mg/dL at 120′ from the load;-A novel diagnosis of diabetes: blood glucose after loading >200 mg/dL.

Renal function was evaluated through serum creatinine values (assay was performed using Jaffe Reaction) and the estimation of glomerular filtration rate (eGFR) using the MDRD formula [10].

Urinary protein excretion was assessed by measuring proteinuria on 24 h urinary collection using the immunoturbidimetric method.

For the determination of intact PTH, we used the ECLIA method (Immuno Assay in ElettroChemiLuminescence, Roche, Basel, Switzerland) using the Modular Analytics E170 analyzer. The measuring range was 1.20–5000 pg/mL. The conversion was pg/mL 0.106 = pmol/L. The normal range was 15–65 pg/mL.

25(OH)D levels were evaluated in serum samples with an enzymatic immunoassay (Kit EIA AC-57FI-Boldon 1evelo-diagnostic system, UK), using a highly specific anti-25(OH)D anti-ovine antibody and enzyme-labeled avidin (horseradish peroxidase). The sensitivity threshold was 5 nmol/mL (2 ng/mL). The specificity of the antiserum was tested with the following analytes calibrated at the level of 50% bond of the zero standard, cross-reactivity: 25-hydroxyvitamin D3 100%; 25-hydroxyvitamin D2 75%; 24,25-Dihydroxyvitamin D3 100%; Cholecalciferol (D3) < 0.01%; Ergocalciferol (D2) > 0.30%. Intra-assay accuracy was calculated from 10 duplicate determinations of two samples each, performed in a single test (CV between 5.3% and 6.7%). The inter-assay accuracy was calculated from duplicate determinations of two samples performed in 11 assays (CV between 4.6% and 8.7%).

All other biochemical parameters were dosed according to the routine methodology used in our central laboratory.

### 2.3. Follow-Up

Patients were followed up until 5 May 2020 and for a mean time of follow-up of 103 ± 60 months. Graft loss and death with functioning graft were considered as the main clinical outcomes. Glycemic status was monitored until T12.

Patients lost at the follow-up during the reference interval of the study (n = 18) were not considered in the study.

### 2.4. Statistical Analyses

The statistical analysis was carried out using the IBM SPSS Statistics for Mac, version 27 (IBM Corp., Armonk, NY, USA). Continuous variables were expressed as mean ± standard deviation (SD) and, in the presence of variables in which the normal distribution of values was not conserved, as a median [25–75th percentile]. The differences between the various groups were determined using the Student *t*-test, the Wilcoxon–Mann–Whitney test, the χ^2^ test, and the Fisher test where indicated. For the univariate and multivariate analysis, linear and logistic regression models, respectively, were utilized. Comparative analyses were carried out between the ND and D groups, ND and PTDM+, and AMG− and AMG+. In the evaluation of clinical outcomes, a comparative evaluation was also carried out between the AMG− group and the private AMG+ group of patients with frank diabetes. Survival analyses were performed by means of Kaplan–Meier models taking as a reference the log rank significance test. In all analyses carried out, statistical significance was set for *p* < 0.05, with two-tailed values.

## 3. Results

### 3.1. Characteristics of the Cohort Studied and Comparison between Groups

A total of 832 KTxps were enrolled in the study. The characteristics of all patients are shown in Table 1 and Table 2. The mean age was 49 ± 13 years, with a mean dialysis vintage of 54 ± 52 months. Hemodialysis was the therapy before KTx in 70.6% of patients, whereas 20.9% had been treated with peritoneal dialysis. A pre-emptive KTx was received by 8.5% of patients. Fifty-seven percent of the patients were male, and 83.5% had a KTx from a deceased donor.

Both at T1 and T12, immunosuppressive therapy was mainly composed of steroids (average cumulative dose at T12: 2914 ± 962 mg), calcineurin inhibitors, and mycophenolate (Appendix A).

In our cohort at T12, the number of diabetic patients (D) was 237 (28.6%), of which 51 (6.2%) were already affected by diabetes before KTx. Of note, we diagnosed diabetes by OGTT in 23 KTxps, while in 90 KTxps, OGTT showed prediabetic alterations.

### 3.2. Comparison between D and ND Patients

D patients, compared to the 595 ND, were significantly older (*p* < 0.001). In group D, the male gender was significantly more prevalent than the female (63.7% vs. 36.3%, *p* = 0.030). In addition, 65% of D had a positive medical history of hemodialysis (*p* = 0.022) and positive family anamnesis for diabetes (*p* = 0.007). Diabetic patients also had significantly worse renal function indices from the beginning of the observation (eGFR at T1 *p* = 0.003, T6 and T12 *p* < 0.001, creatinine at T12 *p* = 0.036), as well as a significantly higher BMI (*p* < 0.001) at 1 and 12 months (Figure 2 and Appendix A. Proteinuria values were significantly higher in group D at T6 (*p* = 0.029) and T12 (*p* = 0.001), while significance was borderline at T1 (*p* = 0.065) (Figure 2 and Appendix A). Concerning lipid metabolism, D patients had significantly higher triglycerides (at T1 *p* < 0.001, at T6 *p* = 0.002, at T12 *p* < 0.001) and total cholesterol (at T6 *p* = 0.004, at T12 *p* = 0.009) than the ND group as well as a significant reduction in HDL cholesterol at T12 (*p* = 0.020); markers of lipid metabolism showed a decreasing trend in both groups during observation (Figure 2 and Appendix A). Concerning inflammatory parameters, Group D had significantly higher CRP values (at T1 *p* = 0.002, at T6 *p* = 0.008, at T12 *p* < 0.001). Furthermore, we noted poorer blood pressure regulation, particularly regarding systolic blood pressure, in D patients (at T6 *p* = 0.009, at T12 *p* = 0.022) (Figure 2 and Appendix A). There were no discernible differences in hemoglobin, uric acid, albumin, and mineral metabolism parameters. These factors remained significant even twelve months post-KTx (Appendix A).

### 3.3. Comparison between PTDM and ND Patients

The comparative examination of patients who developed diabetes after kidney transplantation (PTDM+) and those without diabetes (ND) revealed a higher age at the time of transplantation (*p* < 0.001) as well as a higher body mass index (BMI) (*p* < 0.001) in the first group (Figure 3 and Appendix A). Furthermore, hemodialytic treatment was reported more frequently in PTDM+ patients (in 15.8%, *p* = 0.041). PTDM+ patients also had worse renal function parameters: eGFR (at T1 *p* = 0.012, at T6 *p* < 0.001, at T12 *p* = 0.001) and creatinine (at T6 *p* = 0.050, at T12 *p* = 0.042), along with increased proteinuria (T12 *p* = 0.006) (Figure 3 and Appendix A).

PTDM+ patients had significantly higher systolic blood pressure at T6 (*p* = 0.006) and T12 (*p* = 0.013) (Figure 3 and Appendix A).

In addition, they had worse metabolic and inflammation markers both at baseline and during FU, with higher levels of cholesterol (*p* < 0.001), triglycerides (*p* < 0.001), and uric acid at T12 (*p* = 0.050) and CRP (at T1 and T6 *p* = 0.002, at T12 *p* < 0.001) (Figure 3 and Appendix A). No other differences in biochemical characteristics were observed between the two groups (Appendix A).

In the multivariate analysis (Table 3), by logistic regression considering the development of PTDM as a dependent variable, age at transplantation (*p* < 0.001), BMI at T1 (*p* = 0.002), and triglycerides at T1 (*p* = 0.010) were independently related to the development of PTD.

### 3.4. OGTT Analysis: Comparison between AMG+ and AMG−

OGTT was performed in 484 KTxps. Among them, abnormalities of glucose metabolism were found in 113 KTxps, 23 of which showed indices compatible with diabetes mellitus, while 90 demonstrated indices compatible with IFG or IGT. For this comparative analysis, patients who were shown to be diabetic in the OGTT were included in the AMG+ group (Appendix A). AMG + patients were older (*p* < 0.001) and had a significantly higher BMI at T1 (*p* = 0.001) and T12 (*p* = 0.009) than AMG− patients *(*Figure 4 and Appendix A). In addition, in this case, AMG+ patients had significant alterations in lipid metabolism such as hypertriglyceridemia (at T6 *p* = 0.001 and T12 *p* = 0.006) and a reduction in HDL at T12 (*p* = 0.024) as well as significantly worse renal function parameters, both in terms of serum creatinine (at T1 *p* = 0.015 and T12 *p* = 0.001) and in terms of eGFR (at T1 *p* = 0.004, at T6 *p* = 0.002, at T12 *p* = 0.004) (Figure 4 and Appendix A). Proteinuria values were significantly higher only at T12 (*p* = 0.045), not reaching significance at either T1 or T6 (Figure 4 and Appendix A).

Again, AMG+ patients had significantly higher CRP values (*p* = 0.032 at T1, *p* < 0.001 at T6 and T12) maintaining significance at T1, T6, and T12 (Figure 4 and Appendix A).

Concerning mineral metabolism, PTH levels were significantly higher in AMG+ group patients at the three time points considered (*p* = 0.055 at T1, *p* = 0.020 at T6, *p* = 0.011 at T12), while 25(OH)D was lower only at T1 (*p* = 0.022) (Figure 4 and Appendix A). At T12, significantly higher values of SBP (*p* = 0.002), DBP (*p* = 0.023), and uric acid (*p* = 0.031) were found (Figure 4 and Appendix A).

In the multivariate analysis, age at transplantation (*p* = 0.006), 25(OH)D (*p* = 0.005), and BMI (*p* = 0.027) at T1 and triglycerides at T6 (*p* = 0.021) were the factors that were most related and independently related to glucose metabolism alterations in the OGTT (Table 4).

### 3.5. Differences in Glucose Metabolism Parameters among the Groups

The total cohort (Table 2): Among the 832 KTx studied for this work, the main glucose metabolism parameters at T1, T6, and T12 were analyzed. In particular, the total cohort had blood glucose levels of 89 ± 23 mg/dL, 92 ± 24 mg/dL, and 89 ± 24 mg/dL at T1, T6, and T12 respectively. The median insulin values were 8.70 [6.30–12.15] μIU/mL, 8.20 [5.90–11.79] μIU/mL, and 8.60 [5.90–11.45] μIU/mL at T1, T6, and T12, respectively. Finally, glycated hemoglobin levels at the three time points were 36 ± 8 mmol/mol, 39 ± 8 mmol/mol, and 40 ± 8 mmol/mol.

D vs. ND (Figure 2 and Appendix A): From the statistical analyses, D had blood glucose values (*p* < 0.001) and glycated hemoglobin (*p* < 0.001) significantly more altered than the ND population. Insulinemia, on the other hand, was substantially overlapping between the two groups and far from achieving statistical significance.

PTDM + vs. ND (Figure 3 and Appendix A): PTDM+ patients had significantly higher insulinemia values at T1 (*p* = 0.035). At T6 and T12, however, there were no statistically significant differences between the two groups. Glycated hemoglobin was higher at the three time points considered (*p* < 0.001) in the PTDM+ group, as was blood glucose (*p* < 0.001).

AMG + vs. AMG− (Figure 4 and Appendix A): Compared to patients who had no alterations in glucose metabolism in the OGTT, AMG+ patients had significantly higher blood glucose and glycated hemoglobin levels at T1, T6, and T12 (*p* < 0.001), while their T6 insulinemia levels were borderline (*p* = 0.053).

### 3.6. Impact of Glucose Metabolism Alterations on Long-Term Clinical Outcome

During the follow-up time, 90 KTx (10.8%) restarted dialysis.

Survival analyses comparing ND and D groups (Figure 5A) demonstrated no statistical significance regarding graft survival. Similarly, when the same type of analysis was performed considering the ND group and the PTDM+ group, the development of PTD was not found to affect the survival of the graft in the long term (Figure 5B).

A very similar result was obtained when comparing the AMG+ group with the AMG− group, considering the AMG+ group both in its entirety and excluding patients diagnosed with diabetes mellitus by the OGTT (Figure 5C,D).

In our study, we also analyzed the impact of alterations in glucose metabolism on KTxps survival with functioning grafts. Specifically, during the follow-up time, 80 KTxps (9.6%) died. In this case, survival analysis demonstrated significantly worse long-term survival for KTxps with diabetes (Figure 6A). On the other hand, when we considered the comparison between the ND group and the PTDM+ group for the same event, the survival analysis showed no significant differences between the two groups compared to the death event with a functioning graft (Figure 6B). Instead, very interesting results emerged from the comparison between the AMG+ vs. AMG− groups. A worse survival curve in the AMG+ group (*p* < 0.001), both considered as a whole and deprived of patients who tested positive for diabetes mellitus in the OGTT, was found (Figure 6C,D).

## 4. Discussion

Diabetic nephropathy is one of the most relevant causes of ESRD, and KTx is the best therapeutic option for this condition [11].

The first objective of this study was to assess the prevalence of diabetic patients who had access to KTx in our center between January 2004 and October 2020. According to the results, the prevalence of 6.2% in our center is considerably lower than that reported in the literature. In any case, it is still significantly increasing in recent years, considering that the prevalence relative to the decade 2004–2014 alone was 2.7%.

This study focused on meticulously analyzing the onset of PTDM. Our findings, indicating a PTDM development rate of 22.4% one year post-KTx between 2004 and 2022, fall within the wide range reported in the existing literature. Consistently, a US multicenter study conducted by Malik RF et al. also reported a comparable PTDM rate of 21.5% post-KTx one year after KTx [12]. Differently, Xu J. et al.’s research in China and Lima C. et al.’s study in Brazil yielded higher values, whereas a Korean single-center study involving 1080 patients reported a substantially lower percentage (11.8%) compared to the former findings [13,14,15]. This great variability is due to the strong differences in uses and customs between different countries and centers regarding individual predisposition to the development of diabetes, immunosuppressive therapeutic protocols, and diagnostic parameters.

Particular attention was also reserved for the study of anamnestic, biochemical, and clinical parameters, in an attempt to identify those most related to the development of PTDM. The age at transplantation is certainly a prominent risk factor for PTDM development, as consistently documented across various studies in the literature [16].

High BMI was also related to the onset of PTDM among the patients we considered, and total cholesterol and triglycerides, in fact, were significantly higher, at the three time points studied, in PTDM patients.

The pathophysiological mechanism appears to be linked to the pro-inflammatory impact on adipose tissue in obese individuals. The literature consistently supports the association between metabolic factors and the risk of developing PTDM [17,18,19].

In addition, an association between inflammation and PTDM development is reported in the literature [20]. In accordance, a correlation between inflammatory status and PTDM was found in our cohort.

Patients with PTDM had lower eGFR levels than non-diabetic patients from the first observation. At the same time, creatinine values were significantly higher at T6 and T12, and proteinuria values were only significantly higher at the last time point considered. It is known in the literature that the presence of alterations in glucose metabolism can determine an impairment of renal function, but also that an altered renal function can act on glucose metabolism, especially in terms of increased insulin resistance [21]. Our results totally confirm this evidence.

Several studies show that immunosuppressive therapy is one of the main risk factors for developing PTDM. In particular, the greater diabetogenic power of tacrolimus compared with cyclosporine and the considerable impact of corticosteroids in relation to cumulative doses emerge. mTOR inhibitor drugs seem to lead to an increased risk, intermediate between that of cyclosporine and that of tacrolimus. As for antiproliferative agents such as mycophenolate and azathioprine, no significant correlations with diabetes have emerged [22]. It is important to underscore that none of the time points considered steroid therapy, expressed in terms of average exposure to steroids, was significantly correlated to diabetes. This result contrasts partially with what was reported in our previous work. The reason can be related to the fact that transplant patient programs in our department, especially in the last 5 years, are designed to avoid and possibly treat glycemic complications from the first moments after transplantation. In addition, the possibility of choosing from a large number of immunosuppressive drugs allows the prescription of increasingly personalized therapeutic regimens.

Also, the greater diabetogenic power of tacrolimus compared with cyclosporine was not confirmed by our analysis. This is probably because only a small number of patients were treated with ciclosporin (7–8%) in our cohort. Arguably for the same reason, no associations with an increased risk of developing PTDM were found for mTOR inhibitor drugs.

In line with what has been reported in the literature, no link has been found between mycophenolate and PTDM+.

Six months after the transplant, we performed the OGTT for 484 KTxps. The low prevalence of PTDM diagnosed with this method compared to the total one is explained by considering that most of the diagnoses of PTDM occurred in the first six months after KTx, when the inflammation and steroid therapy are stronger, and that patients already known for diabetes were obviously not tested.

Older age was found to be a determining factor for the development of abnormalities in glucose metabolism. The results were comparable to the previous comparison (ND vs. PTDM+) also regarding renal function and BMI parameters.

Concerning lipid metabolism, it is possible to highlight how triglycerides, also in this case, have significantly higher levels among patients with altered glucose metabolism in which significantly lower levels of HDL cholesterol have also been detected at T6 and T12. T6 triglycerides and T1 BMI retained their significance even in multivariate analysis.

Interesting results have also emerged regarding mineral metabolism. A 2013 meta-analysis highlighted that higher values of circulating 25OH vitamin D were related to a lower risk of developing type 2 diabetes mellitus [23]. Awena Le Fur et al. confirmed this data with a study carried out on 444 patients with renal transplantation in which they observed that above all, the marked deficit of 25(OH)D understood as values below 10 ng/mL was a factor of independent risk for PTDM [24].

Although 25(OH)D did not significantly influence the comparison between the ND and PTDM groups, in the comparison between the AMG+ and AMG− groups, it was significantly and independently related to the development of AMG at the time of the OGTT.

In support of the numerous studies on the importance of inflammation in the development of insulin resistance and alterations in glucose metabolism, at the three time points considered, we found significantly higher levels of PTH and PCR in patients belonging to the AMG+ group.

Finally, to establish the long-term impact of diabetes mellitus, PTD, and AMG on key clinical outcomes, as clinical outcomes, we mainly considered the loss of the graft and the death of the patient with a functioning kidney.

In survival analyses, no statistical significance emerged in graft loss, regardless of the comparison between groups made: ND vs. D, ND vs. PTDM+, and AMG+ vs. AMG−.

The motivation could be in the fact that the restart of dialysis is not a parameter that can be determined objectively but is defined by numerous variables that relate to the physician’s choices in relation to the patient’s clinical status and future prospects. In addition, the quality of the transplanted organ and the numerous immunological factors that may occur during transplantation may also have a negative impact on a possible correlation.

This result is in line with a recent meta-analysis published in 2019 in *CJASN* in which, with a high quality of evidence, no statistically significant association between diabetes and graft loss at one year was observed. Similarly, ethnicity, BMI, and pre-transplant hypertension do not seem to have a particular effect. The parameters that are significantly correlated are instead the age of donor and recipient, HLA mismatches, and transplantation from a deceased donor compared to that from a living donor [25].

Using the same curves, we analyzed the impact of alterations in glucose metabolism on the survival of patients belonging to the various groups considered. Mortality was higher in the whole diabetic group than in the non-diabetic group. However, the data were not confirmed by considering patients who had developed PTDM and had a mortality rate comparable to that of ND. This finding can be related to the careful follow-up to which patients are subjected and the prompt setting up of an antidiabetic therapy able to prevent the complications most commonly related to patient death. Effectively, many studies are available in the literature to support our findings. Gaynor J. J. et al., in a randomized controlled trial, observed that while pre-transplant diabetes plays a significant role in reducing the life expectancy of the transplanted patient, PTDM does not appear to have an unfavorable impact on either patient or graft survival [26]. Recently, Ünlütürk et al. reported that PTDM had no significant effect on all-cause mortality in 703 KTxps, in agreement with our result. Hussain A. et al., in an observational study published in 2022, showed how pre-transplant diabetes is related to high mortality, while PTD, at least if diagnosed within 5 years of follow-up, does not seem to be associated with it [27].

The most significant finding was observed in the comparison between the AMG− and AMG+ groups, where Kaplan–Meier analyses demonstrated a notable decrease in patient survival among those in the AMG+ group, highlighting the potential long-term negative impact of glucose metabolism alterations that do not meet the criteria for diabetes. Another point that this result allows us to emphasize is the importance of the OGTT, which is able to provide a complete overview of a patient’s glucose metabolism even in the absence of obvious alterations or in the presence of sub-optimal blood glucose values. Even the use of the test as a pre-transplant screening, perhaps in patients with particular characteristics, might be considered by all centers.

This work, including a substantial number of patients, has allowed us to have a broad overview of the alterations in glucose metabolism in KTxps. A further strength of the study is the monocentric design, so the patients in follow-up could have been studied and treated on all aspects using common parameters and uniform treatments.

Nevertheless, the retrospective design of the study and the fact that the OGTT test for clinical and logistical–organizational reasons was not performed in all suitable patients could be considered a limit. In addition, our findings cannot be generalized to other populations of transplant patients or to the general populations, and the study design did not permit the correction of potential confounders.

## 5. Conclusions

In conclusion, the prevalence of diabetic patients who accessed KTx at our department, in the period between January 2014 and October 2020, was found to be a total of 6.2%, significantly lower than that reported in the literature. The prevalence of patients who had developed PTDM one year after transplantation was 22.4%.

We found numerous clinical, biochemical, and anthropometric parameters that proved to be significantly related to the development of diabetes mellitus and alterations in glucose metabolism in general. In particular, the advanced age at transplantation, parameters indicative of altered lipid metabolism, and high inflammatory indices seemed the most relevant. Finally, we established how the alterations in glucose metabolism had no impact on the loss of the KTx. However, comparing the total number of diabetic patients with non-diabetic ones, diabetics had higher mortality. These data were not confirmed by comparing only PTDM+ patients with non-diabetic ones. Alterations in glucose metabolism detected in the OGTT were also correlated with an increase in the mortality of patients with functioning kidneys. This allowed us to highlight the impact of prediabetes status and the importance of OGTT as a screening test in post-transplant and potentially pre-transplant patients. In addition, this last result could raise awareness not only about the importance of routinely performing OGTT after KTx, but also about comprehensive monitoring and modification of factors determining mortality risk in these patients (such as a sedentary lifestyle and balanced diet). Our results confirm what has been previously observed in the literature, contributing to reducing the variability of available evidence in this field. This is made possible by the extensive case analysis of patients from a unique transplant center, which yields reliable and translatable data.

This study helps lay the foundations for future prospective randomized work in which it will be interesting to introduce the use of new antidiabetic drugs with a protective effect at the renal level to study their effects on kidney transplant patients.

## Figures and Tables

**Figure 1 nutrients-16-01520-f001:**
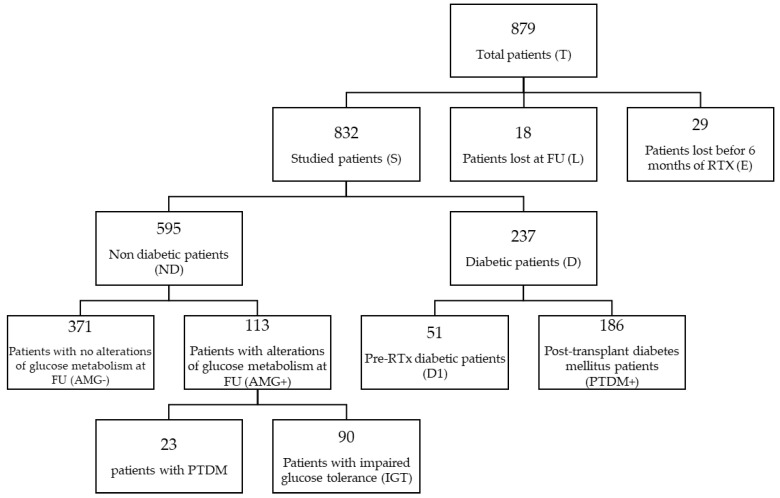
Patient selection and group allocation chart. Note: FU: follow-up.

**Figure 2 nutrients-16-01520-f002:**
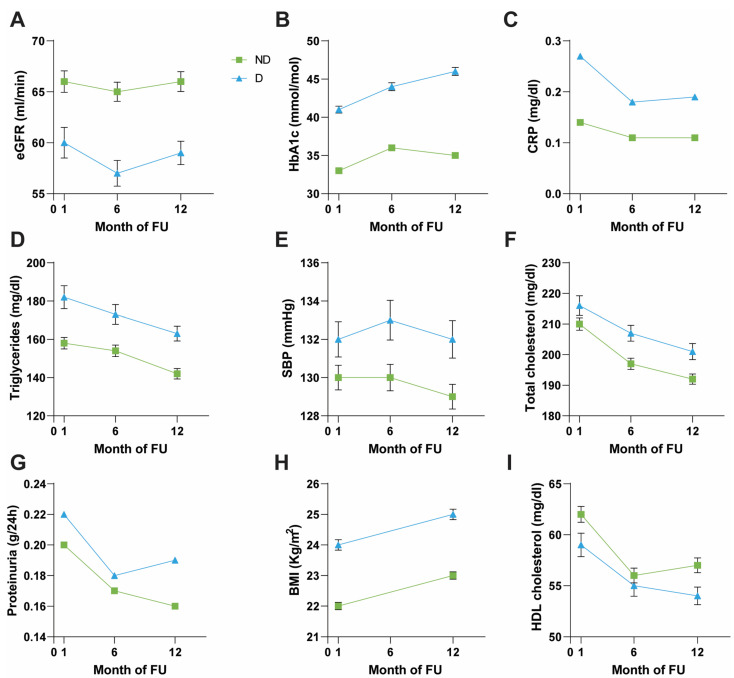
Comparison of clinical and biochemical parameters between ND and D groups. ND: non-diabetic (green line); D: diabetic (blue line). (**A**) eGFR: estimated glomerular filtration rate. (**B**) HbA1c: glycated hemoglobin. (**C**) CRP: C-reactive protein. (**D**) Triglycerides. (**E**) SBP: systolic blood pressure. (**F**) Total cholesterol. (**G**) Proteinuria. (**H**) BMI: body mass index. (**I**) HDL cholesterol.

**Figure 3 nutrients-16-01520-f003:**
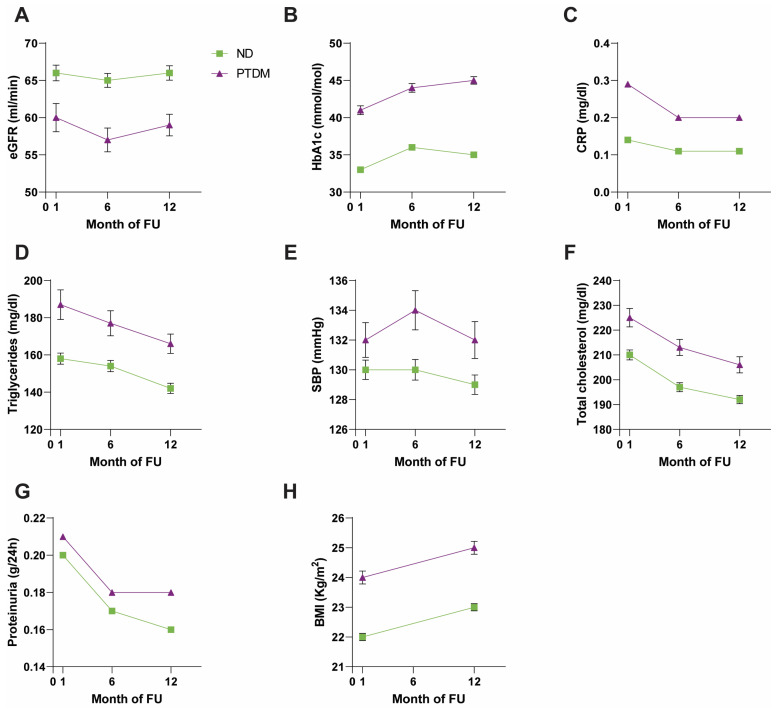
Comparison of clinical and biochemical parameters between ND and PTDM groups. ND: non-diabetic (green line); PTDM: post-transplant diabetes mellitus (violet line). (**A**) eGFR: estimated glomerular filtration rate. (**B**) HbA1c: glycated hemoglobin. (**C**) CRP: C-reactive protein. (**D**) Triglycerides. (**E**) SBP: systolic blood pressure. (**F**) Total cholesterol. (**G**) Proteinuria. (**H**) BMI: body mass index.

**Figure 4 nutrients-16-01520-f004:**
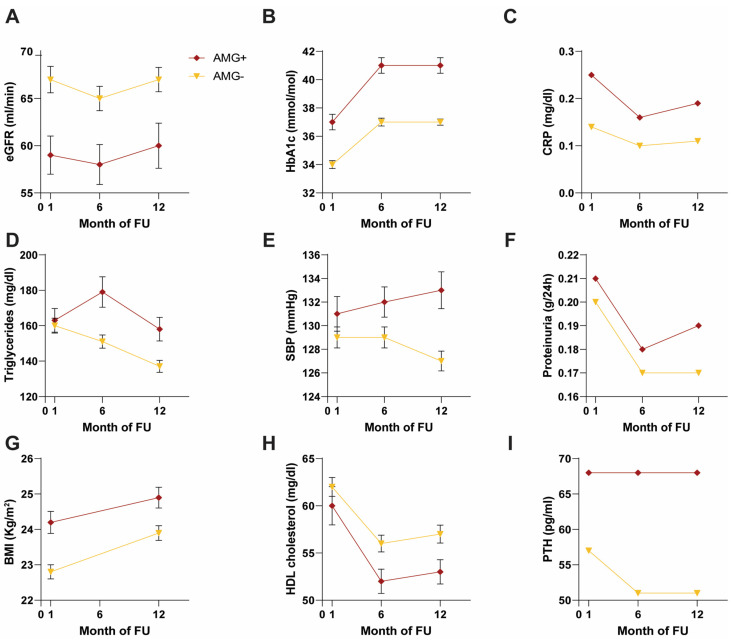
Comparison of clinical and biochemical parameters between AMG− (yellow line) and AMG+ (red line) groups. ND: non-diabetic; D: diabetic. (**A**) eGFR: estimated glomerular filtration rate. (**B**) HbA1c: glycated hemoglobin. (**C**) CRP: C-reactive protein. (**D**) Triglycerides. (**E**) SBP: systolic blood pressure. (**F**) Proteinuria. (**G**) BMI: body mass index. (**H**) HDL: cholesterol; (**I**) PTH: parathoromone.

**Figure 5 nutrients-16-01520-f005:**
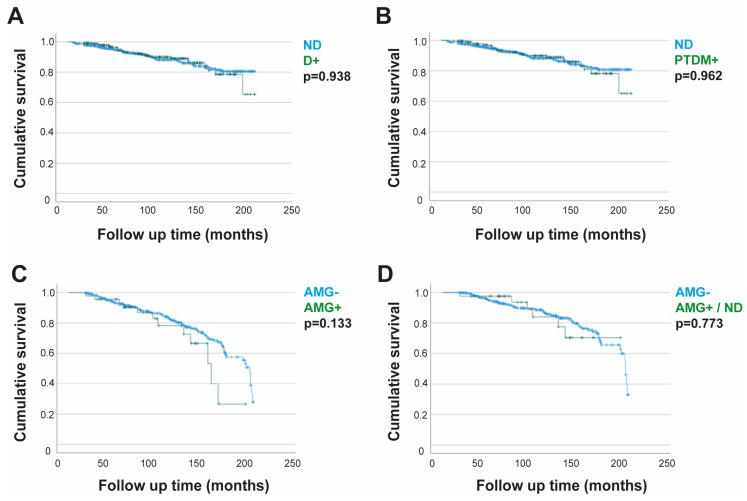
Kaplan–Meier survival curve for the dialysis re-entry event. (**A**) ND and D groups. (**B**) ND and PTDM+ groups. (**C**) AMG− and AMG+ groups. (**D**) AMG− and AMG+ groups without diabetes. Notes: ND: non-diabetic; D: diabetic; PTDM: post-transplant diabetes mellitus; AMG−: no alterations in glucose metabolism; AMG+: alterations in glucose metabolism.

**Figure 6 nutrients-16-01520-f006:**
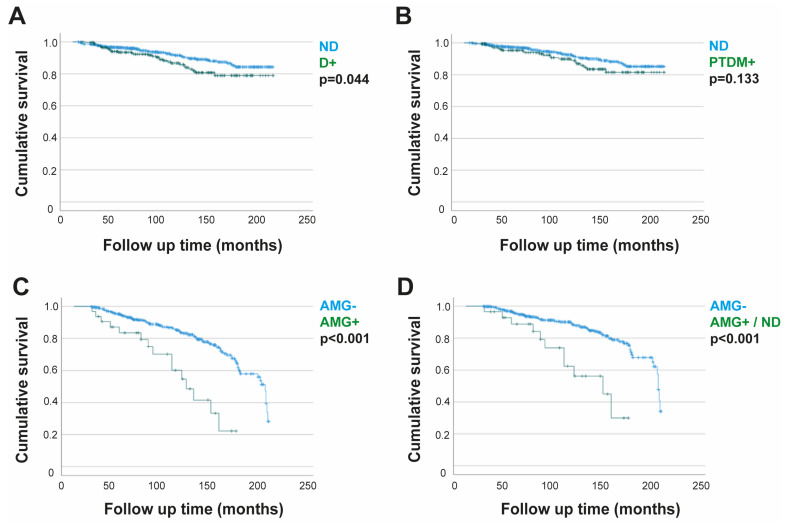
Kaplan–Meier survival curve for the death event. (**A**) ND and D groups. (**B**) ND and PTDM+ groups. (**C**) AMG− and AMG+ groups. (**D**) AMG− and AMG+ groups without diabetes. Notes: ND: non-diabetic; D: diabetic; PTDM: post-transplant diabetes mellitus; AMG−: no alterations in glucose metabolism; AMG+: alterations in glucose metabolism.

**Table 1 nutrients-16-01520-t001:** Features of the cohort divided into groups; notes: * = non-diabetic (ND) vs. diabetic (D), ^ = non-diabetic (ND) vs. post-transplant diabetes mellitus (PTDM+), ° = alterations in glucose metabolism (AMG+) vs. no alterations in glucose metabolism (AMG−); HD: hemodialysis; PD: peritoneal dialysis; KTx: renal transplant. Significant *p*-values are in bold.

Parameters	T (n = 832)	ND (n = 595)	D (n = 237)	PTDM+ (n = 186)	AMG+ (n = 113)	AMG− (n = 371)	*p*
**Age at transplantation, years**	49 ± 13	47 ± 13	54 ± 11	53 ± 11	53 ± 12	46 ± 13	*** <0.001** **^ <0.001** **° <0.001**
**Gender (females/males)**, **%**	42.2% 57.8%	44.5% 55.5%	36.3% 63.7%	43.6% 56.4%	35.4% 64.6%	44.2% 55.8%	*** 0.030** ^ 0.149 ° 0.081
**Type of dialysis** **(no/HD/ PD),** **%**	8.5% 70.6% 20.9%	8.7% 72.9% 18.4%	8.0% 65.0% 27.0%	8.1% 65.0% 26.9%	5.3% 70.8% 23.9%	9.4% 66.3% 24.3%	*** 0.022** **^ 0.041** ° 0.384
**Dialysis vintage,** **months**	54 ± 52	55 ± 53	50 ± 48	51 ± 47	54 ± 47	51 ± 50	* 0.163 ^ 0.365 ° 0.654
**Type of transplant (deceased/living),** **%**	83.7% 16.3%	82.5% 17.5%	86.9% 13.1%	84.9% 15.1%	88.4% 11.6%	79.0% 21.0%	* 0.120 ^ 0.369 ° **0.027**
**Familiar history of diabetes,** **%**	23.0%	20.2%	30.0%	24.5%	24.0%	20.4%	*** 0.007** ^ 0.314 ° 0.267
**Steroid therapy before KTx,** **%**	39.2%	40.6%	35.7%	37.0%	32.1%	39.6%	* 0.222 ^ 0.423 ° 0.176

**Table 2 nutrients-16-01520-t002:** Clinical and biochemical parameters of the overall cohort; notes: eGFR: estimated glomerular filtration rate; BMI: body mass index; SBP: systolic blood pressure; DBP: diastolic blood pressure; PTH: parathormone; Ca: calcium; P: phosphorus; Mg: magnesium; CRP: C-reactive protein; 25OH(D): 25-OH vitamin D; 1-25OH(D): 1-25-OH vitamin D.

Parameters	1st Month (T1)	6th Month (T6)	12th Month (T12)
eGFR (mL/min)	64.66 ± 26.71	63.00 ± 23.70	64.27 ± 23.66
Creatinine (mg/dL)	1.45 ± 0.57	1.64 ± 5.72	1.41 ± 0.45
Proteinuria (g/24 h)	0.20 [0.14-0.30]	0.18 [0.11–0.26]	0.17 [0.11–0.26]
BMI (kg/m^2^)	23.42 ± 3.75	-	24.32 ± 3.85
SBP (mmHg)	130.62 ± 16.45	131.23 ± 17.52	129.91 ± 16.80
DBP (mmHg)	79.71 ± 10.39	79.94 ± 10.28	79.63 ± 10.21
Uric acid (mg/dL)	5.79 ± 1.59	6.51 ± 1.52	6.54 ± 1.55
Hemoglobin (g/dL)	10.98 ± 1.36	12.29 ± 1.48	12.76 ± 1.60
Albumin (g/dL)	4.14 ± 0.43	4.40 ± 0.36	4.40 ± 0.36
PTH (pg/mL)	63 [38–104]	57 [38–94]	55 [37–88]
Ca (mg/dL)	9.74 ± 0.79	9.84 ± 0.72	9.82 ± 0.70
P (mg/dL)	2.54 ± 0.90	3.10 ± 0.70	3.13 ± 0.64
Mg (mg/dL)	1.63 ± 0.23	1.74 ± 0.22	1.71 ± 0.23
Glucose (mg/dL)	88 ± 23	91 ± 23	89 ± 23
Insulinemia (μIU/mL)	8.70 [6.30–12.15]	8.20 [5.90–11.79]	8.60 [5.90–11.45]
Glycated hemoglobin (mmol/mol)	36.42 ± 7.69	39.31 ± 7.74	39.76 ± 8.29
Alkaline phosphatase (U/L)	88 [68–121]	91 [67–125]	83 [62–111]
Total cholesterol (mg/dL)	212 ± 50	200 ± 45	195 ± 43
HDL cholesterol (mg/dL)	61 ± 19	55 ± 17	56 ± 17
Triglycerides (mg/dL)	165 ± 85	159 ± 80	148 ± 68
CRP (mg/dL)	0.17 [0.08–0.54]	0.12 [0.07–0.30]	0.13 [0.07–0.34]
25OH(D) (ng/dL)	14.43 ± 7.93	16.70 ± 10.04	19.24 ± 11.56
1-25OH(D) (ng/dL)	38.33 ± 23.83	49.09 ± 20.82	52.44 ± 20.82

**Table 3 nutrients-16-01520-t003:** Multivariate analysis between ND and PTDM+ groups; notes: ND: non-diabetic; PTDM+: post-transplant diabetes mellitus; BMI: body mass index; significant *p*-values are in bold.

Parameters	*p*	Odds Ratio	*CI*	
**Age at transplantation**	**<0.001**	1.032	1.016	1.048
**BMI T1**	**0.002**	1.080	1.029	1.133
**Total cholesterol (mg/dL) T1**	0.126	1.003	0.999	1.007
**Triglycerides (mg/dL) T1**	**0.010**	1.003	1.001	1.005
**Hemodialysis**	0.268	0.688	0.356	1.332
**Peritoneal dialysis**	0.876	1.059	0.518	2.161

**Table 4 nutrients-16-01520-t004:** Multivariate analysis between the AMG− and AMG+ groups. Notes: AMG−: no alterations in glucose metabolism; MG+: alterations in glucose metabolism; BMI: body mass index; CRP: C-reactive protein. Significant *p*-values are in bold.

Parameters	*p*	Odds Ratio	*IC*	
**Age at transplantation**	**0.006**	1.034	1.010	1.059
**BMI T1**	**0.027**	1.091	1.010	1.178
**25-OH (ng/mL) T1**	**0.005**	0.935	0.892	0.979
**Triglycerides (mg/dL) T6**	**0.021**	1.004	1.001	1.007
**CRP (mg/dL) T1**	0.380	1.047	0.945	1.159
**Type of transplant**	0.325	1.526	0.658	3.540

## Data Availability

The original contributions presented in the study are included in the article, further inquiries can be directed to the corresponding author.

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
