# Peer review of "Post-Transplant Diabetes Mellitus in Kidney-Transplanted Patients: Related Factors and Impact on Long-Term Outcome"

_nutrients, 2024, doi:10.3390/nu16101520_

Round 1

Reviewer 1 Report

Comments and Suggestions for Authors

I have reviewed the manuscript "post transplant diabetes mellitus in kidney transplanted patients: factors related and impact on long-term outcome", which concerns relevant data to investigate determinants of glucose metabolism abnormalities and their impact on long-term clinical outcomes. 

It seems to me that the results do not truly bring something novel to current literature: 

- "Patients with glucose metabolism abnormalities were older, exhibited altered lipid profiles, higher Body Mass Index, and increased inflammatory indices" --> known

- "Mortality risk was associated with pre-transplant diabetes and glucose metabolism abnormalities" --> known

Thus I would suggest to adapt the proposal of the manuscript, results, interpretation and future perspectives under such consideration.

"PTDM within the first year" seems to used with the same meaning and at times with a different meaning than "glucose metabolism abnormalities detected via oral glucose tolerance test". Definition of PTDM should be crystal clear. All definitions should be supported by references. Terminology needs to be used consistently throughout the manuscript. 

- I could not find reporting of patients lost to follow up. How were they handled?

- Please add a patient flowchart, starting with the number of patients eligible for recruitment. 

- Tables and figures both need serious editing. It is not actually possible to look properly at the Figures, I could not judge them in its current version. 

Comments on the Quality of English Language

Moderate english language and style revision needed.

Author Response

ANSWERS TO REVIEWER 1

We thank the reviewer for all his suggestions. We appreciate the insightful comments made by the reviewer. In response, we made our best efforts to improve the manuscript. We would signal that all the manuscript has been reviewed for the English grammar by a mother-tongue English teacher.

  • - "Patients with glucose metabolism abnormalities were older, exhibited altered lipid profiles, higher Body Mass Index, and increased inflammatory indices" --> known

- "Mortality risk was associated with pre-transplant diabetes and glucose metabolism abnormalities" --> known

Thus I would suggest to adapt the proposal of the manuscript, results, interpretation and future perspectives under such consideration.

We thank the reviewer for the suggestion, we amended the text in discussion

  • "PTDM within the first year" seems to used with the same meaning and at times with a different meaning than "glucose metabolism abnormalities detected via oral glucose tolerance test". Definition of PTDM should be crystal clear. All definitions should be supported by references. Terminology needs to be used consistently throughout the manuscript. 

We thank the reviewer for this comment. Now all the definitions are correctly reported in the text with reference.

  • I could not find reporting of patients lost to follow up. How were they handled?

We thank the reviewer for his comment: as now reported in the text, patients lost at the follow up during the reference interval of study (n= 18) were not considered in the study.

  • Please add a patient flowchart, starting with the number of patients eligible for recruitment. 

We thank the reviewer for the suggestion, we added a flowchart in the main text to better clarify our criteria for patients' selection.

  • Tables and figures both need serious editing. It is not actually possible to look properly at the Figures, I could not judge them in its current version.

We thank the reviewer for the suggestion. We edited and implemented both tables and figures format to improve the quality of the layout. 

Reviewer 2 Report

Comments and Suggestions for Authors

This study is intriguing as it seeks to explore the frequency and factors influencing irregularities in glucose metabolism, as well as their consequences on the extended clinical prognosis among individuals who have undergone kidney transplantation. However, certain amendments should be made to improve the paper. First and foremost, the value of the Abstract should be improved by highlighting more relevant results and pertaining p-values. In the Introduction section, when it is stated "Consequently, to make a diagnosis of PTDM, it is advisable to wait for the stabilization of the patient's immunosuppressive therapy and renal function and to ascertain the absence of concomitant infectious events", specific infectious events should be mentioned. Some abbreviations are mentioned, but not defined (such as PTDM). Methodological part of the manuscript should be backed up with a flowchart for easier understanding. The authors should also provide more context or justification for selecting specific parameters measured. More context should be provided on the study location (what is the caption area of this polyclinic, the average number of patients, etc.). Software used should be properly cited: "IBM SPSS Statistics for Windows, version 27 (IBM Corp., Armonk, N.Y., USA". It should be stated whether p-value was two-tailed. The authors should ensure consistency in formatting and presentation throughout the Discussion, as often single sentences are used for a whole paragraph (which detracts from main messages of the paper). Study limitations should be addressed in greater depth (with an emphasis on low generalizability of the findings to other populations or settings, failure to control for confounders that could impact the validity of the study's conclusions, measurement errors, etc). The Discussion section should include more newer references (as the ones from 2023 are lacking). Conclusion section should propose potential avenues for future research. The whole manuscript should be additionally proofread by a native English speaker.

Comments on the Quality of English Language

It is advisable to have the manuscript undergo further proofreading by a proficient English speaker to ensure linguistic accuracy and clarity.

Author Response

ANSWERS TO REVIEWER 2

We thank the reviewer for all his suggestions. We appreciate the insightful comments made by the reviewer. In response, we made our best efforts to improve the manuscript. We would signal that all the manuscript has been reviewed for the English grammar by a mother-tongue English teacher.

This study is intriguing as it seeks to explore the frequency and factors influencing irregularities in glucose metabolism, as well as their consequences on the extended clinical prognosis among individuals who have undergone kidney transplantation. However, certain amendments should be made to improve the paper.

  • First and foremost, the value of the Abstract should be improved by highlighting more relevant results and pertaining p-values.

We thank the reviewer for his comment. Unfortunately, was not simple to summarize all the study design and the main results in 200 words. In any case, a global revision of the abstract taking in account of the suggestion was made.

  • In the Introduction section, when it is stated "Consequently, to make a diagnosis of PTDM, it is advisable to wait for the stabilization of the patient's immunosuppressive therapy and renal function and to ascertain the absence of concomitant infectious events", specific infectious events should be mentioned.

We thank the reviewer for the suggestion. We have completed the sentence according the reviewer suggestion

  • Some abbreviations are mentioned, but not defined (such as PTDM).

We thank the reviewer for the remark and we are sorry for the mistake. We amended the main text defining every abbreviation.

  • Methodological part of the manuscript should be backed up with a flowchart for easier understanding.

We thank the reviewer for the suggestion, we added a flowchart in the main text to better clarify our criteria for patients' selection.

  • The authors should also provide more context or justification for selecting specific parameters measured.

We thank the reviewer or the suggestion. This part was added in the materials and methods

  • More context should be provided on the study location (what is the caption area of this polyclinic, the average number of patients, etc.).

We thank the reviewer for the suggestion. We amended the main text better defining our study location.

  • Software used should be properly cited: "IBM SPSS Statistics for Windows, version 27 (IBM Corp., Armonk, N.Y., USA". It should be stated whether p-value was two-tailed.

We thank the reviewer for the suggestion. Now all these details are reported in the main text.

  • The authors should ensure consistency in formatting and presentation throughout the Discussion, as often single sentences are used for a whole paragraph (which detracts from main messages of the paper).

Answer: A global revision of the discussion was performed.

  • Study limitations should be addressed in greater depth (with an emphasis on low generalizability of the findings to other populations or settings, failure to control for confounders that could impact the validity of the study's conclusions, measurement errors, etc).

ANSWER: We totally agree with the reviewers, so we added this point in the discussion.

  •  
  • The Discussion section should include more newer references (as the ones from 2023 are lacking). Conclusion section should propose potential avenues for future research.

ANSWER: we thank the reviewer for his suggestion, and we added some recent publications about the topic.

Round 2

Reviewer 1 Report

Comments and Suggestions for Authors

This manuscript concerns relevant data to investigate determinants of glucose metabolism abnormalities, yet it needs to be further underscored that I once more fail to acknowledge what this manuscript actually provides in addition to what is already well-known and established. The data would be expected to be very useful but I fail to recognize what is value can we take from going through the entire manuscript. It is hard for me to justify all these data and tables to the readership only leading to the conclusion that nothing can be added.  

- in the revised version of the manuscript I could not find appropriate support to justify the method used to handle lost to follow up patients. I did not find either discussion of potential bias introduced by this method. 

- in the revised version of the manuscript, once again I did not find information on the starting number of patients eligible for recruitment, unless indeed 100% of elegible patients were actually included (with exception of those thereafter excluded by lost to FU). If the latter is the case, it should be clearly stated.

Comments on the Quality of English Language

To my best consideration, the manuscript concerns unfinished work because with all that data novelty at least by providing a different approach to the analyses is missing. It is hard for me to justify all these data and tables to the readership only leading to the conclusion that nothing can be added.  

Author Response

ANSWERS TO REVIEWER 2

We thank the reviewer for all his suggestions. We appreciate the insightful comments made by the reviewer. In response, we made our best efforts to improve the manuscript.

This manuscript concerns relevant data to investigate determinants of glucose metabolism abnormalities, yet it needs to be further underscored that I once more fail to acknowledge what this manuscript actually provides in addition to what is already well-known and established. The data would be expected to be very useful but I fail to recognize what is value can we take from going through the entire manuscript. It is hard for me to justify all these data and tables to the readership only leading to the conclusion that nothing can be added. 

Answer: we thank the reviewer for his comment. In our opinion, beyond providing a monocentric overview of a large number of patients with the issue, our study's main and intriguing result is the role of OGTT in identifying abnormalities in glucose metabolism, which have themselves been correlated, regardless of the presence of diabetes, with mortality. Therefore, our study could raise awareness not only about the importance of routinely performing OGTT but also about closely monitoring those patients who show abnormalities in glucose metabolism regardless of diabetes. We further emphasized this point in the conclusions of the work, relating it even more closely to the objective of the special issue for which it was proposed.

- in the revised version of the manuscript I could not find appropriate support to justify the method used to handle lost to follow up patients. I did not find either discussion of potential bias introduced by this method.

Answer: The reviewer's observation is entirely accurate, and we sincerely apologize for this oversight. Indeed, the patients lost to follow-up were inadvertently excluded from the global cohort from the outset. We have now rectified this error in both the text and the figures. It's important to note that this adjustment does not introduce any bias into the analysis; rather, these patients were excluded from the cohort from the beginning.

- in the revised version of the manuscript, once again I did not find information on the starting number of patients eligible for recruitment, unless indeed 100% of elegible patients were actually included (with exception of those thereafter excluded by lost to FU). If the latter is the case, it should be clearly stated.

Answer: We thank the reviewer for this point. Now we re-edited the sentence in which is reported the number of patients according to the global elegible cohort (inclunding the 29 that did not reach the sixth month of follow-up and the 18 that were lost during the global follow up): 879.

Comments on the Quality of English Language

To my best consideration, the manuscript concerns unfinished work because with all that data novelty at least by providing a different approach to the analyses is missing. It is hard for me to justify all these data and tables to the readership only leading to the conclusion that nothing can be added.

ANSWER: We thank the reviewer for his comment. We believe that beyond the discussed data points, the strength of our study lies in its large sample size within a single-center setting. This is significant for ensuring consistency in managing the studied cohort, mitigating potential biases often present in multicenter studies. While it's true that our tables were extensive and somewhat challenging to navigate, we included numerous variables to provide a comprehensive depiction of the studied cohort. Following recommendations from the academic editor and reviewer 2, we opted to include only significant variables in the tables. However, all variables, including those that did not show statistical significance, will be documented in the supplementary materials.

Round 3

Reviewer 1 Report

Comments and Suggestions for Authors

no more comments

Comments on the Quality of English Language

no more comments